# Efficiency Assessment of Bacterial Cellulose on Lowering Lipid Levels In Vitro and Improving Lipid Metabolism In Vivo

**DOI:** 10.3390/molecules27113495

**Published:** 2022-05-30

**Authors:** Wen Zhang, Qian-Yu Zhang, Jian-Jun Wang, Le-Le Zhang, Zhen-Zhen Dong

**Affiliations:** 1School of Food and Biological Engineering, Shaanxi University of Science and Technology, Xi’an 710021, China; zqy15181651630@163.com (Q.-Y.Z.); 15288169957@163.com (J.-J.W.); 19829260572@163.com (L.-L.Z.); 2Shaanxi Xi’an Lantian County Food and Drug Inspection Testing Center, Xi’an 710500, China; dongdoublez@163.com

**Keywords:** bacterial cellulose, hyperlipidemia mice, lipid metabolism, intestinal microflora

## Abstract

Bacterial cellulose (BC) is well known as a high-performance dietary fiber. This study investigates the adsorption capacity of BC for cholesterol, sodium cholate, unsaturated oil, and heavy metal ions in vitro. Further, a hyperlipidemia mouse model was constructed to investigate the effects of BC on lipid metabolism, antioxidant levels, and intestinal microflora. The results showed that the maximum adsorption capacities of BC for cholesterol, sodium cholate, Pb^2+^ and Cr^6+^ were 11.910, 16.149, 238.337, 1.525 and 1.809 mg/g, respectively. Additionally, BC reduced the blood lipid levels, regulated the peroxide levels, and ameliorated the liver injury in hyperlipidemia mice. Analysis of the intestinal flora revealed that BC improved the bacterial community of intestinal microflora in hyperlipidemia mice. It was found that the abundance of *Bacteroidetes* was increased, while the abundance of *Firmicutes* and *Proteobacteria* was decreased at the phylum level. In addition, increased abundance of *Lactobacillus* and decreased abundance of *Lachnospiraceae* and *Prevotellaceae* were obtained at the genus level. These changes were supposed to be beneficial to the activities of intestinal microflora. To conclude, the findings prove the role of BC in improving lipid metabolism in hyperlipidemia mice and provide a theoretical basis for the utilization of BC in functional food.

## 1. Introduction

Since the beginning of the 21st century, obesity has become a global public health problem, due to changes in diet structure and living standards [1]. Epidemiological studies have shown that obesity is a significant risk factor for diabetes, cardiovascular diseases, cancer, hypertension, hyperlipidemia, and premature death [2,3]. Some countries have implemented public health strategies to prevent hyperlipidemia, obesity, and diabetes through diet regulation [4,5,6]. Over the past few decades, dietary fiber has attracted attention due to various pharmacological and clinical applications. Studies have shown significant positive effects of increased consumption of dietary fiber in reducing obesity [7], colon cancer [8], cardiovascular diseases, and hyperlipidemia [9]. The increased intake of dietary fiber from grains, vegetables, and fruits reduced the risk of obesity and hyperlipidemia [3,7,10,11,12,13]. Dietary fiber significantly inhibited the increase in serum levels of total lipids (TL), triglycerides (TG), total cholesterol (TC), and low-density lipoprotein cholesterol (LDL-C) [7,14,15], which promoted the increase in high-density lipoprotein cholesterol (HDL-C) level as well as the excretion of fecal cholesterol and fecal bile acids in mice [12,13,15,16]. Brockman et al. [17] found that high-viscosity dietary fiber effectively reduced obesity and fatty liver degeneration in rats fed with a high-fat diet. Huang et al. [18] proved the capacity of soybean dietary fiber to adsorb cholesterol and bile acids. Luo et al. [12] discovered that dietary fiber extracted from bamboo shoot shell bound with fat, cholesterol, bile acids, and nitrite, which ameliorated lipid metabolism disorder of hyperlipidemia mice. Zhang et al. [19] investigated the ability of water-soluble dietary fiber extracted from apple peel, wheat bran, and bean. This study reported that the adsorption capacity of soluble dietary fiber for toxic metal ions such as lead (Pb^2+^), mercury (Hg^+^), and cadmium (Cd^2+^) varied under different pH conditions. At neutral pH, close to the intestinal pH, water-soluble dietary fiber strongly bound with toxic metal ions. In addition, dietary fiber played an important role in regulating intestinal microflora [16]. Turnbaugh et al. [20] found that a diet with low fat and high dietary fiber positively regulated the intestinal microflora. Meanwhile, Cheng et al. [16] showed that dietary fiber increased intestinal probiotics, promoted hepatic and intestinal circulation of lipid metabolism, and facilitated lipid (cholesterol) transformation and excretion. 

Bacterial cellulose (BC) is a long-chain, high-molecular-weight polymer synthesized by microorganisms. It consists of glucopyranosyl units linked by *β*-1,4 glycosidic bond and has a unique three-dimensional structure composed of nanoscale microfiber. Moreover, BC contains a large number of hydrophilic groups such as hydroxyl, which provides the possibility to bind other substances [21]. BC is a high-quality dietary fiber widely used in food, biomedical, and chemical industries [21,22,23,24]. Numerous studies have shown that BC has good adsorption properties. Spherical BC, prepared by Zhu et al. [25], adsorbed bovine serum albumin (BSA) and was recovered from BSA-BC complex. BC molecules absorbed metals ions from wastewater due to hydroxyl groups contributing to high specific surface area [26]. Xu et al. [27] used BC as the substrate to construct molecularly imprinted materials, which displayed high selectivity towards a mixture of structural analogues, including p-cresol isomers (o-cresol, m-cresol, and p-cresol). BC membranes adsorbed non-metallic toxins such as histamine, ammonia, and NO^2−^ [28]. The meatballs made using BC to replace a portion of fat presented the characteristics of high fiber and low fat and calorie, achieving the purpose of a healthy diet [29,30]. However, there are few systematic studies on the fat reduction effects of BC. 

This study aimed to evaluate the abilities of BC to reduce lipids in vitro and improve the lipid metabolism of hyperlipidemia mice. In vitro, the adsorption capacity of BC on lipid metabolism-related substances (cholesterol, sodium choline, unsaturated fat and heavy metal ions) were explored. In vivo, BC was added at high, middle, and low doses to a high-fat diet to carry out the random intervention feeding experiment and evaluated the effects of BC on blood lipid index, liver histopathology, and intestinal microflora in hyperlipidemia mice. This study will provide a theoretical basis for the utilization of BC in functional foods.

## 2. Materials and Methods

### 2.1. Materials and Chemicals

BC was produced using *Acetobacter xylinum* (CGMCC 1.1812) by static fermentation, following the method described by W. Zhang [24]. Specific-pathogen-free (SPF) Kunming male mice (6–8 weeks, 18–22 g) and the basic diet were purchased from Shaanxi Medical Experimental Animal Center (Shaanxi, China). Cholesterol and bile acids were purchased from Shanghai Macklin Biochemical Co., Ltd. (Shanghai, China). Lovastatin was purchased from Shanghai Yuanye Bio-Technology Co., Ltd. (Shanghai, China). The kits used to determine lipase, total bile acids, HDL-C, LDL-C, total cholesterol (T-CHO), TG, superoxide dismutase (SOD; WST-1 method), alanine aminotransferase (ALT/GPT; ELISA), and aspartate aminotransferase (AST/GOT) were obtained from Nanjing Jiancheng Bioengineering Institute (Nanjing, China). Other biochemical reagents were obtained locally, and all reagents were of analytical grade.

### 2.2. In Vitro Adsorption Properties of BC

The adsorption of cholesterol, sodium cholate, unsaturated oil, and heavy metal ions were determined by the reported method with slight modifications [12,18,19,26,31,32,33]. Cholesterol content in the egg yolk emulsion was determined at 550 nm by o-phthalaldehyde method in ultraviolet-visible spectrophotometer (Beijing Purkinje General Instrument Co., Ltd., Beijing, China). Sodium cholate content was determined at 620 nm by furfural colorimetry in U-Vis. The concentration of unsaturated oil was determined at 530 nm following the vanillin phosphate chromogenic method in U-Vis. The mass concentration of heavy metal ions in the solution was determined by the atomic absorption spectrophotometry in the Z-2000 atomic absorption spectrometer (Hitachi Ltd., Tokyo, Japan).

During the adsorption experiments, different suspensions were prepared by dispersing BC powder into different aqueous media. The pH of the media was adjusted using hydrochloric acid solution or sodium hydroxide solution. The details of the suspensions were listed in Table 1. The prepared suspensions were cultured at 37 °C for 3 h in a shaking incubator at 100 r/min. At 15, 30, 45, 60, 90, 120, 150, and 180 min, the suspensions were withdrawn, and the concentrations of cholesterol, sodium cholate, unsaturated oil and heavy metal ions were determined based on the methods described previously.

The adsorption quantity and adsorption rate of BC for cholesterol, sodium cholate, unsaturated oil, heavy metal ions were calculated using Equations (1) and (2). The adsorption curve and adsorption rate curve of BC for cholesterol were constructed as a function of time.
(1)$$Q=\frac{C-C_{e}}{M}\times V$$
where

*Q*—the adsorption quantity of cholesterol, sodium cholate, unsaturated oil and heavy metal ions adsorbed onto BC (mg/g);

*C*—the cholesterol, sodium cholate, unsaturated oil and heavy metal ions content in system before adsorption (mg/mL);

*C_e_*—the cholesterol, sodium cholate, unsaturated oil and heavy metal ions content in system after adsorption (mg/mL);

*V*—the volume of suspensions (mL);

*M*—BC mass (g).
(2)$$\frac{dQ}{dt}=\frac{d\left(\frac{C-C_{e}}{M}\times V\right)}{dt}$$
where

*dQ/dt*—the adsorption rate of BC for cholesterol, sodium cholate, unsaturated oil and heavy metal ions (mg/(g·min));

*t*—adsorption time (min).

### 2.3. The Effect of BC on the Improvement of Lipid Metabolism in Hyperlipidemia Mice

#### 2.3.1. Establishment of a Hyperlipidemia Mouse Model and Feeding Interventions

The use of animals was conducted in compliance with the European guidelines for the care and use of laboratory animals. The study was approved by the Experimental Animal Center of Xi’an Jiaotong University (Xi’an, China) whose approval license number is SCXK-(Shaan) 2017-003. Experimental protocols were approved by the Institutional Animal Care and Use Committee of Xi’an Jiaotong University Health Science Center (No. 2020-078) in compliance with the animal management rules of the Chinese Ministry of Health.School of Food and Biological Engineering, Shaanxi University of Science and Technology, China. The high-fat diet was composed of 42% basic diet, 22% cooked lard, 24% white granulated sugar, 9% egg yolk, and 3% edible salt. Freeze-dried BC membrane was crushed and sieved through a 100-mesh sieve, and 1%, 5%, and 10% (weight) of this were added to the high-fat diet to prepare a low-dose, middle-dose, and high-dose BC feed, respectively. Seventy SPF-grade Kunming male mice were given adaptive feeding for 10 days under a relative humidity of 50%, temperature of 23 °C, and light and dark cycle of 12/12 h. The mice were randomly divided into seven groups, such as basic diet control group (BG), high-fat model group (MG), positive control group (PG), negative control group (NG), low-dose BC group (LDG), middle-dose BC group (MDG), and high-dose BC group (HDG). Six groups, except BG, were fed with a high-fat diet to establish the hyperlipidemia model. Six weeks later, three mice were randomly selected from each group fed with high-fat diet for orbital blood collection, and the blood lipid indexes such as TC and TG were measured to determine the success of the model. Then, NG was fed with basic diet, while LDG, MDG, and HDG were fed with low-dose, middle-dose, and high-dose BC feed, respectively, for 8 weeks. The PG group was given lovastatin (0.5 mL/50 g body weight) every afternoon by gavage. During the feeding period, the mice had free access to water and food. The food and water intake, mental state, and hair of the mice were observed and recorded every day. Three days before the end of the experiment, the feces of mice in MG, PG, NG, LDG, MDG, and HDG were collected, sealed, and stored at −80 °C. The mice were fasted for 12 h before the end of the experiment and were sacrificed post anesthesia. Blood was collected from the orbital veins and centrifuged at 3000× *g* rpm for 10 min to separate the serum. Liver tissues of mice were collected and stored at −80 °C for further analysis.

#### 2.3.2. Measurement of Blood Lipid Index

The TC, TG, LDL-C, HDL-C, ALT, AST, and SOD levels in the serum of mice from every group were measured using the specific kits following the manufacturers’ instructions in a Varioskan Flash full-wavelength scanning multi-function reader (Thermo Fisher Technology Co., Ltd., Shanghai, China).

#### 2.3.3. Histopathological Observation of Liver Tissue

The liver tissue’s wet weight was measured, and the liver coefficient (LC) was determined [34]. The liver tissue was fixed with 4% paraformaldehyde, embedded in conventional paraffin, and sectioned (4–6 μm in thickness). Hematoxylin-eosin (HE) staining [3] and Masson’s trichrome staining [35] were performed, and the morphology of liver cells was observed under an optical microscope.

### 2.4. Analysis of Intestinal Microflora of Mice

#### 2.4.1. DNA Extraction and Amplification

Fecal DNA extraction and sequencing were performed by the Shanghai Majorbio company (Shanghai, China). The data obtained were analyzed on the free online platform of Majorbio (www.majorbio.com) (accessed on 12 January 2022). The total DNA was extracted using E.Z.N.A.^®^ Universal Pathogen Kit (Omega Bio-tek, Norcross, GA, USA) following the manufacturer’s instructions. Nanodrop 2000 was employed to detect the concentration and purity of DNA, and 1% agarose gel electrophoresis was adopted to detect the mass of extracted DNA. The V3-V4 variable region of bacterial 16S rRNA was amplified using 338F (5′ ACTCCTACGGGAG 3′) and 806R (5′ GGACTACHVGGGGTWTCTAAT 3′) primers. The PCR was carried out on an ABI GeneAmp^®^ system 9700 (Thermo Fisher Technology Co., Ltd. Waltham, MA, USA) under the following conditions: predenaturation at 95 °C for 3 min; 27 cycles of denaturation at 95 °C for 30 s, annealing at 55 °C for 30 s, and extension at 72 °C for 30 s; a final extension at 72 °C for 10 s. The amplicons were separated on 2% agarose gel, extracted with AxyPrep DNA Gel Extraction Kit (Axygen Biosciences, Union City, CA, USA), and eluted using Tris-HCl. QuantiFluor-ST (Promega, Madison, WI, USA) was used to detect and quantify the DNA.

#### 2.4.2. Illumina Sequencing and Alpha and Beta Diversity Analysis

Based on the standard operating procedures (SOP) of the Illumina MiSeq platform (Illumina, San Diego, CA, USA), the purified fragments were used to construct a PE2 × 300 library. Miseq PE 300 platform (Illumina) was employed to perform multiplex sequencing, and the original data have been uploaded to the NCBI database. Operational taxonomic units (OTU) with 97% similarity were selected, and the alpha diversity index was calculated using Mothur software with randomly selected samples. The data were visualized using R-chart. We further investigated the similarities and dissimilarities in species composition among the samples. Cluster analysis was carried out on the distance matrix generated by samples, and hierarchical clustering was performed based on the beta diversity distance matrix.

#### 2.4.3. Processing of Sequencing Data

Trimmomatic was used for quality filtering of the raw file in FASTQ format, and the high-quality reads were merged using FLASH. The standards used in data processing were as follows: (1) Filtered the reads with a read-tail mass value less than 20, and then set a window of 50 bp. If the average mass value in the window was lower than 20, the bases at the end of a read were cut off; the reads below 50 bp after mass control were filtered, and the reads with N base were removed. (2) Based on the overlap among PE readers, the paired reads were merged into one sequence with a minimum overlap length of 10 bp. (3) The maximum mismatch ratio allowed in the overlap region of the splicing sequence was 0.2; the mismatched sequences were screened. (4) According to the barcodes and primers at the beginning and end of the sequence, the samples were distinguished, and the sequence direction was adjusted. The maximum number of mismatches allowed for barcode was 0 and for primers was 2. OTU clustering based on 97% similarity was performed using Usearch. The sequences classified as chimera were removed. Based on the 16S rRNA SILVA database (release 128), the RDP classifier algorithm was employed to classify and analyze gene sequences of each 16S rRNA, with a 70% confidence threshold.

### 2.5. Data Processing

All experiments were carried out three times to obtain mean values. Microsoft Excel and Origin 8 software were used to process the data, and Graphpad statistical software was used to perform statistical analysis and determine significant differences. In figures, * indicate significant difference (*p* < 0.05) while ** indicates extremely significant difference (*p* < 0.01).

## 3. Results and Discussion

### 3.1. In Vitro Adsorption Properties of BC

#### 3.1.1. Adsorption of Cholesterol by BC

Cholesterol is a lipid essential for cellular metabolism and tissue function maintenance in animal cells. The adsorption of cholesterol from food mainly occurs in the gastrointestinal tract [12,18]. However, extremely high cholesterol content will lead to an increase in TC, TG, and LDL-C levels and the decrease in HDL-C level in the blood, causing hyperlipidemia [12,18]. The adsorption curves and adsorption rate curves of BC for cholesterol are shown in Figure 1. At pH 2.0, cholesterol adsorption increased rapidly within 15–60 min; the adsorption quantity slowly increased until it attained equilibrium (9.879 mg/g) in 60–150 min. At pH 7.0, the adsorption quantity of BC for cholesterol increased rapidly in 15–120 min, and remained constant until 150–180 min, with a maximum adsorption quantity of 11.910 mg/g. These findings indicate that BC has a good absorption capacity for cholesterol in the stomach (pH 2.0) and small intestine (pH 7.0). In addition, the cholesterol-adsorption capacities were higher at pH 7.0 (simulation intestinal environment) than pH 2.0 (simulation gastric environment). At pH 2.0, there were more hydrogen ions, which rejected the positive charges carried by cholesterol, resulting in the decreased cholesterol-binding capacity of BC [36]. These results indicated that the BC may exhibit stronger capacities to reduce cholesterol concentrations in the small intestine, which were consistent with the results demonstrated in previous studies [37,38,39].

Meanwhile, the adsorption rate curve shows that under the experimental conditions, the adsorption rate (*dQ_C_/dt*) and the adsorption quantity (*Q_C_*) increased rapidly during the first 60 min. During this period, more hydroxyl groups were present on the surface of BC, and the concentration of cholesterol in the egg yolk solution was high. Due to the adsorption quantity of BC, cholesterol molecules formed a monomolecular layer, and BC rapidly adsorbed the molecules diffused out of the egg yolk emulsion [21]. As the adsorption progressed, the empty pores and the free surface area of BC gradually declined. In addition, the difference in concentration between BC and cholesterol and the adsorption rate gradually decreased, and the adsorption became slow. Further, 120 min later, when the cholesterol concentration inside and outside the BC was almost similar, cholesterol molecule diffusion might have stopped. The van der Waals force between cholesterol and BC also gradually weakened [40], and the adsorption rate of BC decreased to the lowest when the adsorption reached equilibrium.

#### 3.1.2. Adsorption of Sodium Cholate by BC

The adsorption of sodium cholate at different concentrations onto BC is shown in Figure 2. The adsorption of 3 g/L sodium cholate onto BC increased rapidly in 15–120 min and remained almost unchanged during 150–180 min, with maximum adsorption of 16.149 mg/g. These findings indicate that BC effectively adsorbs sodium cholate, and the adsorption is concentration dependent.

The adsorption rate curve shows that the adsorption of sodium cholate onto BC increased gradually until equilibrium (Figure 2B). During the initial stage of adsorption (first 100 min), BC rapidly adsorbed free sodium cholate molecules; the diffusion rate of sodium cholate was the largest, and the adsorption quantity rapidly increased with time. Further, 120 min later, the concentration difference between sodium cholate and BC decreased, and the sodium cholate diffusion rate was inhibited. When the concentration of sodium cholate became similar between the sodium cholate solution and BC, the adsorption rate reduced to a minimum, and the adsorption quantity reached a maximum. During rapid adsorption, the large surface area of BC could have helped bind with sodium cholate [41]. With prolonged adsorption, the adsorption sites gradually decreased, and therefore the adsorption reached equilibrium. The pore adsorption of sodium cholate weakened accordingly. After a certain period, the sodium cholate adsorbed onto BC and that in the solution became equal, and no further diffusion took place; the adsorption reached equilibrium.

#### 3.1.3. Adsorption of Unsaturated Oil by BC

In animals, the adsorption of unsaturated oils mainly occurs in the small intestine [12]. Zhao et al. mentioned that insoluble dietary fiber can effectively adsorb lipids and has high oil holding capacity, which can reduce the absorption and utilization of lipids by the body [42]. Wang et al. study showed that insoluble dietary fiber has high water retention capacity and oil binding capacity [41]. The adsorption of unsaturated oil onto BC rapidly increased in 15–120 min and gradually reached equilibrium (238.337 mg/g; Figure 3A), indicating the adsorption capacity of BC for unsaturated oil.

During the adsorption process, the adsorption rate of BC for unsaturated oil gradually declined. The adsorption rate was high during 30–60 min and declined during 60–100 min. The adsorption rate of unsaturated oil remained almost unchanged from 120 min until the end of the adsorption, which may be affected by adsorption equilibrium. During the initial stage, BC rapidly adsorbed free unsaturated oil molecules, and the largest diffusion rate of unsaturated oil molecules (*dQc/dt*) was achieved. During the adsorption process, the concentration difference between unsaturated oil and BC reduced, and the diffusion of unsaturated oil molecules was inhibited. When there was no difference between the unsaturated oil adsorbed by BC and that in the solution, the adsorption rate of unsaturated oil onto BC decreased to the lowest, and the adsorption reached equilibrium. After that, the adsorption rate of unsaturated oil onto BC showed no change until the end of the reaction.

#### 3.1.4. Adsorption of Cr^6+^and Pb^2+^ by BC

The adsorption curve and adsorption rate curve of Cr^6+^ onto BC at different pH are shown in Figure 4A,B. The adsorption of Cr^6+^ onto BC under acidic conditions increased rapidly in the first 80 min and gradually reached equilibrium (1.809 mg/g) after 120 min. Under neutral conditions, the adsorption quantity of BC for Cr^6+^ increased slowly and reached equilibrium (0.304 mg/g) after 150 min. Under acidic conditions, the adsorption rate of Cr^6+^ onto BC remained constant without any decline during 20–80 min and then decreased rapidly within 80–120 min (Figure 4B). After that, almost no change in the adsorption rate was found until the end of the reaction. Under neutral conditions, the adsorption rate of Cr^6+^ onto BC decreased slowly in 20–120 min and then rapidly reduced until equilibrium.

The adsorption curves and adsorption rate curves of Pb^2+^ onto BC at different pH are shown in Figure 4C,D. Under acidic conditions, the adsorption quantity of BC for Pb^2+^ increased rapidly in 20–80 min, and then the adsorption rate rapidly decreased. The inflection point appeared at 120 min. A slight decrease was observed during 150–180 min, and the maximum absorption capability (1.525 mg/g) was gradually achieved. Under neutral conditions, from 20 to 80 min, the amount of Pb^2+^ adsorbed onto BC increased rapidly, while the adsorption rate rapidly decreased. The inflection point appeared at 120 min, and then the adsorption rate gradually reduced to a minimum. The adsorption quantity reached equilibrium (2.925 mg/g).

During the initial stage of adsorption, the adsorption rate of BC for metal ions was the largest. As the adsorption progressed, the difference in concentration between metal ions in the solution and those adsorbed onto BC decreased gradually, the adsorption rate declined, and the adsorption quantity became close to equilibrium. When there were no differences between the metal ions adsorbed by BC and those in the solution, the adsorption rate of BC decreased to a minimum. By this time, many metal ions were embedded on the surface of BC, the electrostatic repulsion between BC and solution increased, and the adsorption reached equilibrium.

The adsorption of heavy metal ions onto BC is mainly caused by the surface-active groups (hydroxyl groups). The mechanism involved surface energy adsorption, electrostatic attraction, and coordination adsorption [43]. Environmental pH is a major factor that influences hydroxyl groups and Cr^6+^and Pb^2+^on BC surface [19]. The hydroxyl groups on BC undergo hydrogenation and dehydrogenation with a change in pH. The adsorption quantity of BC for Cr^6+^ under acidic conditions was greater than that under neutral conditions, probably due to the combined action of hydroxyl groups on BC and pH of the solution. Under low pH, hydroxyl groups easily bind with H^+^ to perform protonation and generate -OH_2_^+^, which combines with heavy metals such as Cr^6+^ [44]. With increase in pH, part of Cr_2_O_7_^2−^ gets converted to CrO_4_^2−^, leading to the decrease in Cr^6+^; therefore, BC’s adsorption quantity decreased. Under extremely low pH, Pb^2+^ and H^+^ compete for the adsorption sites on BC, and limited protonation occurs. This leads to an increase in the electrostatic repulsion force on BC surface; therefore, the amount of Pb^2+^ adsorbed onto BC decreased. With an increase in pH, the hydroxyl groups on the BC surface dehydrogenate, resulting in an increase in negative charges. Thus, the adsorption quantity gradually increased. When pH continued to increase, the removal rate of Pb^2+^ further increased, probably due to the precipitation of metal hydroxides in the solution, instead of the increase in adsorption by BC [43].

### 3.2. Effect of BC on Lipid Metabolism in Hyperlipidemia Mice

#### 3.2.1. In Vivo Lipid-Lowering Effect of BC

Cholesterol is converted into bile acids via enzymatic reaction in hepatocytes, which are pumped into the bile duct by bile salt export pump. Free bile acids are passively reabsorbed in the small and large intestines by diffusion. The conjugated bile acids get actively absorbed into the intestinal mucosal cells through bile acid transporter in the ileum. Few bile acids bind with the ileal bile acid-binding protein and secrete into the portal venous system [31]. The purpose of this study was to investigate the transport states of lipid metabolism under dietary intervention with BC in mice.

The food and water intake of mice in each group are shown in Figure 5. The food intake of mice in BG first decreased and then gradually increased to the maximum by the 5th week, and then remained constant. The food intake of mice in MG, PG, NG, LDG, MDG, and HDG increased gradually with time, and the maximum was observed around the 5th week. Further, food intake decreased slowly. The water intake of mice in MG increased from 1st to 3rd week, and no significant change was observed from 3rd to 5th week. After the 5th week, the water intake gradually decreased. The water intake of mice in PG showed no obvious change with time. The water intake of mice in NG and LDG first reduced and then increased slowly. The water intake of mice in MDG and HDG increased gradually in the first three weeks and then decreased. This may be because the high-fat diet increased exogenous cholesterol intake, leading to difficulty in bile acid synthesis in animals. BC combines with a part of cholesterol and bile salts, thus improving the cholesterol metabolism in mice [17]. BC also absorbs water several times its weight in the large intestine, accelerates intestinal peristalsis, it and promotes body waste transformation into soft feces, which gets excreted easily [23]. Thus, the water and food intake of mice in MDG and HDG improved.

Hyperlipidemia is a condition in which there are elevated levels of TC, TG, and LDL-C in blood, with a lower level of HDL-C. The changes in these components reflect the blood cholesterol content and the overall metabolic status [18]. Lipoproteins bind with cholesterol and transport it in the bloodstream. LDL-C transports cholesterol from the liver to different tissues and cells, while HDL-C transports cholesterol from other tissues and cells to the liver. The increase in LDL-C level will lead to lipid metabolism disorder, slow down blood flow, and increase blood viscosity, resulting in atherosclerosis due to fat deposition on the arterial walls [45]. The blood lipid components of mice in every group are shown in Figure 6A–D. The TC, TG, and LDL-C levels in mice of LDG, MDG, and HDG were significantly lower, while the HDL-C level was significantly higher (*p* < 0.05) compared with those in MG; however, no significant difference was found compared with PG. Meanwhile, significant differences in the serum HDL-C and LDL-C levels were detected between BC intervention group and MG group, which indicate that BC can prevent atherosclerosis.

ALT and AST are two indicators used to detect liver function and are positively correlated to the degree of liver damage [46]. LC, measured as the ratio of the wet weight of organs to unit weight, reflects the pathological changes or dysplasia of organs [44]. The ALT, AST, LC, and SOD levels in mice of every group are shown in Figure 6E–H. The ALT, AST, and LC levels in mice of LDG, MDG, and HDG were significantly lower, while the SOD level was significantly higher (*p* < 0.05) compared with the MG; however, no significant difference was detected compared with PG. These findings indicate that BC can reduce serum ALT and AST levels, ameliorate hyperlipidemia-induced liver damage, regulate the peroxide levels, and demonstrate an antioxidant effect.

#### 3.2.2. Effect of BC on Hepatic Histomorphology

HE staining of mice liver is shown in Figure 7A. Severe hepatic steatosis was observed in mice of MG. Many diffused lipid vacuoles of different sizes were observed in the interior hepatic cells. In NG, inflammatory infiltration and cell fusion were found in the liver tissues. No fat vacuoles were observed in the liver sections of mice in BG and PG, and the liver tissue was normal. BC diet intervention significantly alleviated the degree of hepatic steatosis in mice of LDG, MDG, and HDG; the number of vacuoles decreased, and the inflammatory response of hepatocytes decreased. These observations indicate that BC diet intervention could ameliorate the high-fat diet-induced liver cell damage in mice to a certain extent.

Masson’s staining, which turns collagen fibers to blue and muscle fibers to red, indicates the fiber and inflammatory factors in tissues [35]. The degree of liver cell fibrosis in mice of BG and PG was lower, suggesting less liver injury; however, large fibrosis was detected in mice of LDG (Figure 7B). The symptoms of liver fibrosis in mice of MDG and HDG were less severe than those in MG. Masson’s staining results indicate that BC can ameliorate liver injury in hyperlipidemia mice.

#### 3.2.3. Diversity of Intestinal Microflora

##### Alpha Diversity and Venn Diagram 

The effect of BC on the intestinal microflora of hyperlipidemia mice was further evaluated. The Shannon and Simpson index plots are shown in Figure 8a,b. The microbial community was less diverse in MG than the other groups, with no significant difference among the groups. This may be because the mice in MG were always fed with a high-fat diet. However, the intake of lovastatin and BC by the mice of the treatment group and diet intervention groups might have led to microflora change.

Further, the Venn diagram was used to analyze the bacterial diversity and evaluate the OTU distribution among different samples [15]. Figure 8c shows that 378 OTUs were common among MG, PG, NG, LDG, MDG, and HDG (588, 581, 616, 527, 585, and 643, respectively). In addition, 10 OTUs were specific to MG, 5 to PG, 2 to NG, 3 to LDG, 11 to MDG, and 11 to HDG. Additionally, 414 OTUs were common to PG, LDG, MDG, and HDG, 40 to MG, LDG, MDG, and HDG, 431 to NG, LDG, MDG, and HDG, 466 to LDG and MDG, 497 to LDG and HDG, and 445 to LDG, MDG, and HDG. Together, these findings indicate that BC, a dietary fiber, can regulate the intestinal microflora in hyperlipidemia mice.

##### Diversity of Intestinal Microflora

Intestinal microflora can be classified into three types: symbiotic flora, opportunistic pathogenic flora, and pathogenic flora. The symbiotic flora mainly includes *Firmicutes*, *Bacteroidetes*, and *Proteobacteria* [47]. Studies have shown that the relative abundance of *Bacteroidetes* in the intestine of mice fed with a high-fat diet was lower, and that of *Firmicutes*, *Proteobacteria*, *Butyrate-*producing bacteria, and *Bifidobacteria* were higher. In addition, dietary fiber significantly increased the relative abundance of *Bifidobacteria* in the intestine [47].

In this study, the microflora abundance was statistically analyzed at various taxonomic levels, and the community composition of different groups was visualized using histograms [48]. The composition of communities at the phylum and genus levels is shown in Figure 9. At the phylum level, the abundance of *Bacteroidetes* (0.3153) in MG was lower compared with the other groups, while the abundances of *Firmicutes* (0.6375) and *Proteobacteria* (0.0131) were higher. The abundance of *Bacteroidetes* (0.4085) in PG was significantly higher compared with MG, but *Firmicutes* (0.5267) was lower to a certain extent; however, the abundance of *Proteobacteria* (0.0297) was higher. The abundance of *Bacteroidetes* (0.6097) in NG was higher compared with PG, while the abundances of *Firmicutes* (0.3040) and *Proteobacteria* (0.0201) were lower. The abundances of *Bacteroidetes* (0.4564) and *Proteobacteria* (0.0341) in LDG were higher compared with PG, while that of *Firmicutes* (0.4708) was lower. The abundance of *Bacteroidetes* (0.5825) in MDG was higher than those in PG and LDG, and the abundance of *Firmicutes* (0.3237) was lower. The abundance of *Proteobacteria* (0.0308) was higher compared with PG but lower compared with LDG. The abundance of *Bacteroidetes* (0.5807) in HDG was significantly higher than PG; it was higher to a certain extent than LDG but lower than MDG. The abundance of *Firmicutes* (0.3024) was significantly lower than that in PG and slightly lower than that in LDG and MDG. The abundance of *Proteobacteria* (0.0217) was lower than that in PG, which was significantly lower than that in LDG and MDG. Our findings together indicate that the intestinal microflora at the phylum level significantly improved in the BC intervention group; BC improved the abundance of *Bacteroidetes* and decreased *Firmicutes* and *Proteobacteria*.

To study the intestinal microflora at a lower level, several representative strains, such as *Bioceroides, Lachnospiraceae, Prevotellaceae*, and *Lactobacillus*, were specially selected to explore the community composition of every group. The abundance of *Bicteroides* (0.1879) in MG was lower compared with PG, NG, LDG, and HDG; however, it was higher compared with MDG. The abundance of *Lachnospiraceae* and *Prevotellaceae* (0.4843) in MG was higher than that in the other groups. The abundance of *Lactobacillus* in MG was less compared with PG, NG, and HDG and high compared with LDG and MDG. The relative abundances of *Bicteroides* (0.3901) and *Lactobacillus* (0.0532) were higher in PG than in MG, while the overall abundance of *Lachnospiraceae* and *Prevotellaceae* (0.2688) was lower. In NG, the abundance of *Bioceroides* (0.2432) was lower compared with that in PG, and the overall abundance of *Lachnospiraceae* and *Prevotellaceae* (0.3556) and the abundance of *Lactobacillus* (0.0580) were higher. The abundance of *Bioceroides* (0.3823) in LDG was slightly lower than that in PG; the overall abundance of *Lachnospiraceae* and *Prevotellaceae* (0.2634) and the abundance of *Lactobacillus* (0.0118) were lower. In MDG, the abundances of *Bioceroides* (0.1832) and *Lactobacillus* (0.0109) were lower and the overall abundance of *Lachnospiraceae* and *Prevotellaceae* (0.3579) was higher compared with PG. In HDG, the abundance of *Bioceroides* (0.2030) and the overall abundance of *Lachnospiraceae* and *Prevotellaceae* (0.2452) were lower compared with PG, while the abundance of *Lactobacillus* (0.1789) was significantly higher. The abundance of *Lactobacillus* in HDG was significantly higher compared with PG, while the overall abundance of *Lachnospiraceae* and *Prevotellaceae* was significantly lower. These findings suggest that BC had a beneficial influence on the intestinal microflora, especially at the genus level.

##### Beta Diversity

PCA analysis chart is shown in Figure 10A. The points with different colors or shapes represented the samples of different groups. In the PCA chart, the closer the two sample points are, the more similar the species composition of the two samples [49,50]. The sample points of PG, NG, LDG, MDG, and HDG were close to each other; they were similar to NG but slightly different from PG and significantly different from MG. The results indicated that the intestinal microflora improved in mice of LDG, MDG, and HDG. BC significantly regulated the host intestinal microflora.

Further, PCoA revealed a significant difference between MG and other groups (Figure 10B). PG was farther from MG, indicating obvious differences in microflora between the two groups. This finding indicates the positive role of lovastatin. We detected slight differences among NG, LDG, MDG, and HDG, probably due to the expansion of microbial species at the genus level. *Lactobacillus* showed significant differences among the BC intervention groups, leading to different manifestations and aggregation of the sample points in the PCoA scatter plot.

NMDS is an analytical approach based on the sample distance matrix. NMDS chart with stress <0.2 has a certain explanatory significance [50]. In addition, in the NMDS analysis chart, the sample points of MG were positioned away from the other groups (Figure 10C). The difference in PG sample points may be because of the difference in drug resistance of each mouse and the difference in OTU of the intestinal microbial flora. The OTUs of LDG, MDG, and HDG were aggregated, and several OTUs were different in the genus, probably due to the difference in BC dose or BC intake by each mouse. To conclude, beta diversity analysis indicates that BC positively influences the intestinal microflora in mice to a certain extent.

## 4. Conclusions

We analyzed BC’s adsorption capacity for cholesterol, sodium cholate, unsaturated oil, and heavy metal ions. We further investigated the mechanisms via which BC affects lipid metabolism using the hyperlipidemia mouse model. BC adsorbed cholesterol, sodium cholate, unsaturated oil, and heavy metal ions to a certain extent. Further, the in vitro adsorption experiments revealed BC’s good lipid-lowering effects. In vivo experiments demonstrated that BC significantly regulated the blood lipid level and ameliorated liver injury in hyperlipidemia mice; it also demonstrated an antioxidant effect. Additionally, BC improved the abundance of intestinal microflora in hyperlipidemia mice. The above results showed a conclusion consistent with previous studies. Bamboo dietary fiber had strong cholesterol-adsorption capacity and prebiotic potential [51]; *Hericium erinaceus* dietary fiber has significant blood lipid-lowering effects and improves the lipid metabolism disorder [52]; dietary fiber extracted from deoiled red raspberry pomace has high adsorption capacities of glucose, cholesterol and toxic ions [53]. Thus, the study’s findings provide a theoretical basis for the utilization of BC in functional foods.

## Figures and Tables

**Figure 1 molecules-27-03495-f001:**
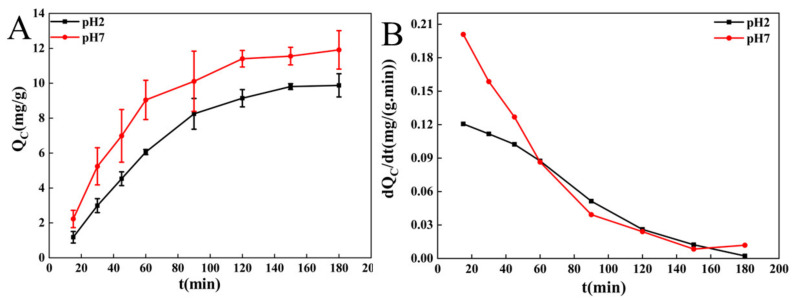
Adsorption curves (**A**) and adsorption speed curves (**B**) of cholesterol onto BC at different pH.

**Figure 2 molecules-27-03495-f002:**
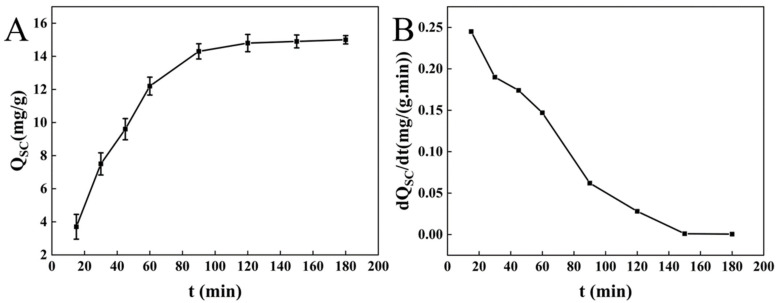
Adsorption curves (**A**) and adsorption speed curves (**B**) of sodium cholate at different concentrations onto BC.

**Figure 3 molecules-27-03495-f003:**
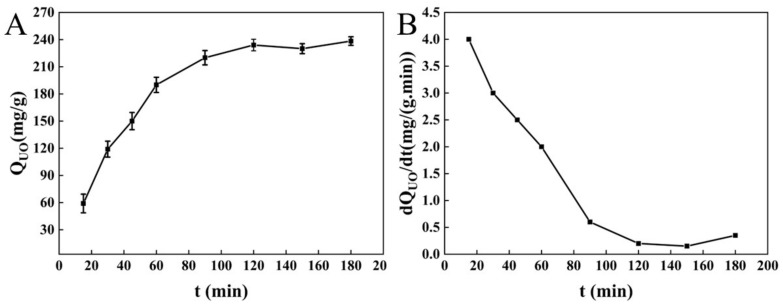
Adsorption curves (**A**) and adsorption speed curves (**B**) of unsaturated oil onto BC.

**Figure 4 molecules-27-03495-f004:**
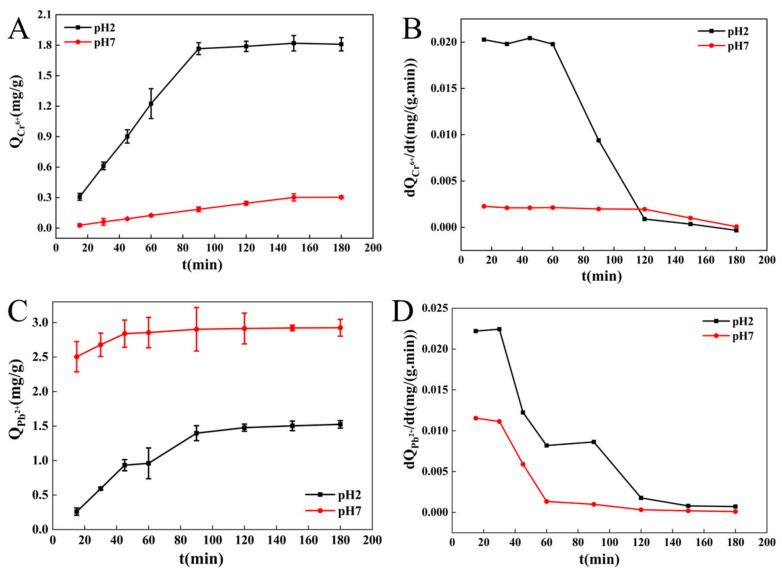
Adsorption of Cr^6+^ (**A**) and Pb^2+^ (**C**) and adsorption speed curves of Cr^6+^ (**B**) and Pb^2+^ (**D**) onto BC at different pH.

**Figure 5 molecules-27-03495-f005:**
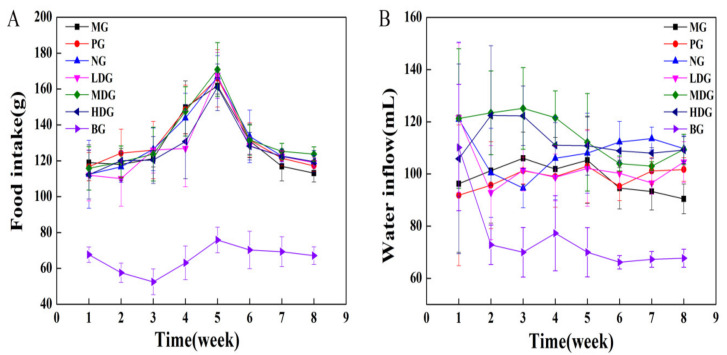
Food (**A**) and water (**B**) intake of mice.

**Figure 6 molecules-27-03495-f006:**
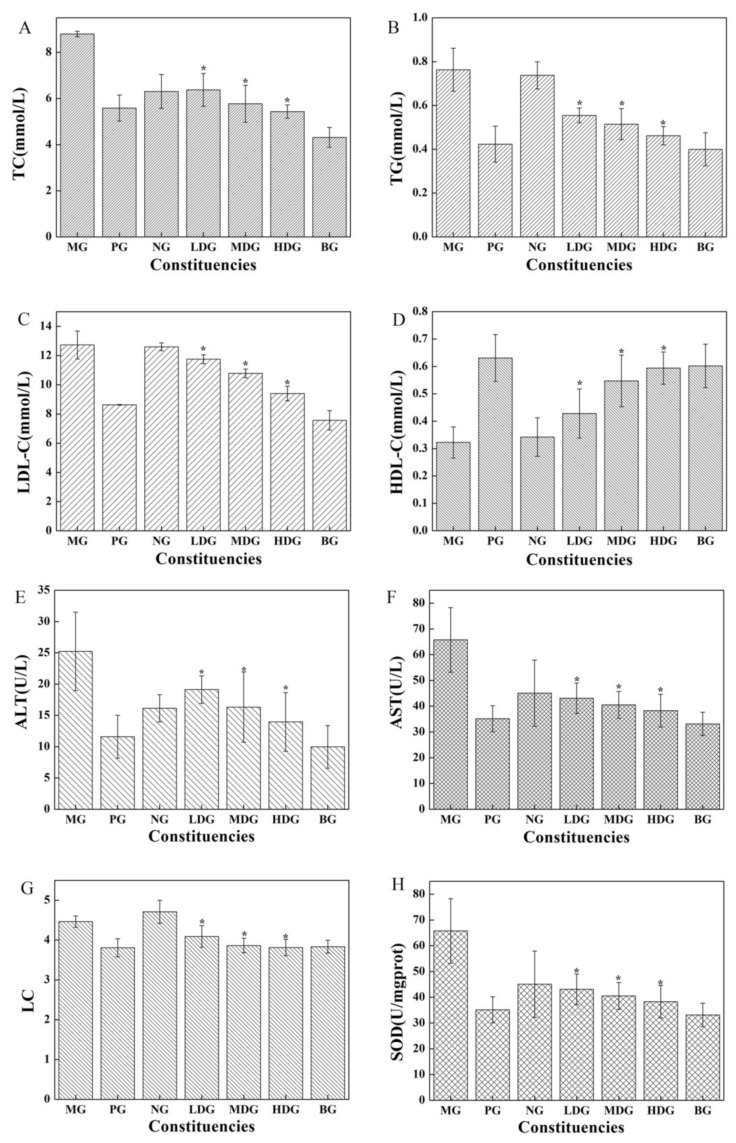
TC (**A**), TG (**B**), LDL-C (**C**), HDL-C (**D**), ALT (**E**), AST (**F**), LC (**G**), and SOD (**H**) levels in mice. * *p* < 0.05 as compared with MG.

**Figure 7 molecules-27-03495-f007:**
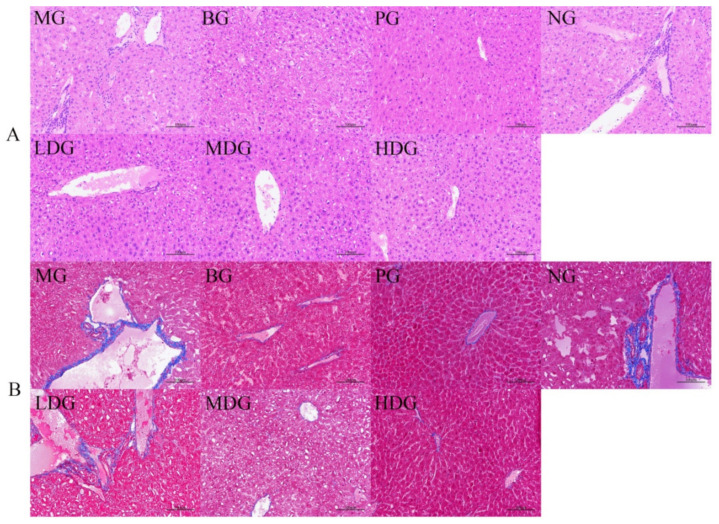
HE staining (**A**) and Masson’s staining (**B**) of liver tissue of mice in each group.

**Figure 8 molecules-27-03495-f008:**
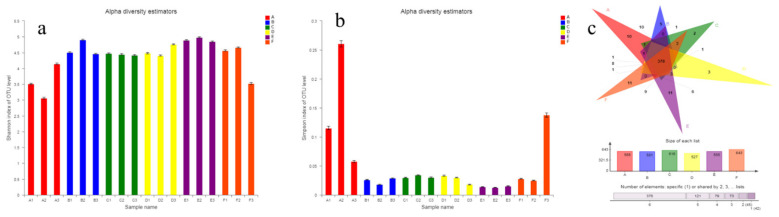
Alpha diversity (**a**,**b**) and Venn diagram (**c**) analysis (A, MG group; B, PG group; C, NG group; D, LDG group; E, MDG group; F, HDG group).

**Figure 9 molecules-27-03495-f009:**
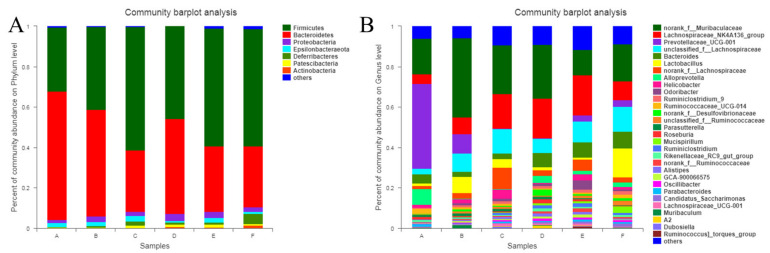
Histogram of the relative abundance of microflora communities at phylum (**A**) and genus (**B**) levels.

**Figure 10 molecules-27-03495-f010:**
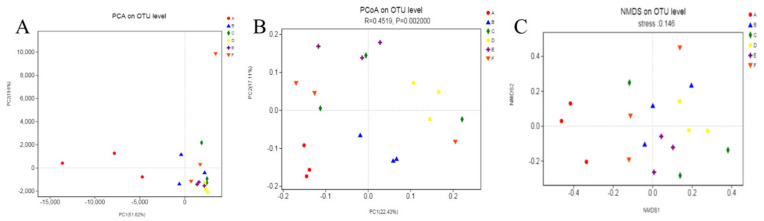
PCA (**A**), PCoA (**B**), and NMDS analysis (**C**) charts.

**Table 1 molecules-27-03495-t001:** The composition of in vitro adsorption experiment (100 mL).

System	BC/g	pH	Egg Yolk Emulsion/mL	Sodium Cholate/g	NaCl/g	Soybean Oil/mL	K_2_Cr_2_O_7_/mg	Pb(NO_3_)_2_/mg	Water/mL
Cholesterol adsorption system	2.0	2.0	10	-	-	-	-	-	90
		7.0							
Sodium cholate adsorption system	2.0	7.0	-	0.3	0.877	-	-	-	100
Unsaturated oil adsorption system	0.8	7.0	-	-	-	100	-	-	-
Cr^6+^ adsorption system	0.08	2.0	-	-		-	0.4	-	100
		7.0			-				
Pb^2+^ adsorption system	0.08	2.0	-	-	-	-	-	0.4	100
		7.0							

## Data Availability

All the data generated during this study are included in this article.

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
