# Peer review of "Efficiency Assessment of Bacterial Cellulose on Lowering Lipid Levels In Vitro and Improving Lipid Metabolism In Vivo"

_molecules, 2022, doi:10.3390/molecules27113495_

Round 1

Reviewer 1 Report

The comments are listed below. 

  1. The title of the manuscript should be modified. The phrase “Effects of bacterial cellulose” implies that bacterial cellulose content is varied to examine how it affects lipid levels.
  2. The experimental design in Table 1 must be explained. For example, why pH was controlled at 7 just for the adsorptions of sodium cholate and unsaturated oil, why only sodium cholate adsorption was studied on the initial concentration effect, and why the BC mass was not controlled at the same value for all adsorption experiments.
  3. A brief of the experimental procedure from the referred paper should be given (lines 77 and 96). 
  4. In Section 2.2, more detail must be provided such as initial concentrations of the substances, total solution volume, etc.
  5. The characteristics of BC powder must be informed and used to explain along with the adsorption mechanism.
  6. The adsorption mechanism must be explained clearly. 
  7. In line 226, the empty pores were mentioned. Were the substances adsorbed inside the BC pores? 
  8. If the adsorption occurs only on the BC surface area, how does diffusion affect adsorption? 
  9. Adsorption type should be classified for each substance. 
  10. The result of the adsorption part was presented without the statistical analysis (the plot without error bars). 
  11. The results should be compared with previous works. 

Author Response

Reviewer 1

1、The title of the manuscript should be modified. The phrase “Effects of bacterial cellulose” implies that bacterial cellulose content is varied to examine how it affects lipid levels.

Response: Thank you for your advice. We have modified the title to “Efficiency assessment of bacterial cellulose on lowering lipid levels in vitro and improving lipid metabolism in vivo”, and marked red in manuscript. (L2)

2、The experimental design in Table 1 must be explained. For example, why pH was controlled at 7 just for the adsorptions of sodium cholate and unsaturated oil, why only sodium cholate adsorption was studied on the initial concentration effect, and why the BC mass was not controlled at the same value for all adsorption experiments.

Response: The adsorption of sodium cholate and unsaturated oil in animals mainly occurs in the small intestine, so only the adsorption at pH=7 was studied.

In order to study the adsorption capacity of BC to sodium cholate, we initially designed the adsorption experiment of BC to sodium cholate at two concentrations. However, due to the same adsorption properties of BC to sodium cholate at these two concentrations, we delete the data about BC to 2g/L sodium cholate solution. And corresponding modifications were made in the manuscript.

About the BC mass was not controlled at the same value for adsorption experiments. After the preliminary experiment, we found that the adsorption capacity of BC to each adsorbent was quite different, in order to control the adsorption capacity at the same level, so different amounts of BC in the formal experiment.

3、A brief of the experimental procedure from the referred paper should be given (lines 77 and 96). 

Response: The modification has been made according to your suggestion, and the revised part has been marked in red. (L82). BC was produced using Acetobacter xylinum (CGMCC 1.1812) by static fermentation, following the method described by W. Zhang.

I'm so sorry that our manuscript was not detailed enough. We have modified in manuscript and marked in red (L94-102). The adsorption of cholesterol, sodium cholate, unsaturated oil, heavy metal ions and the determination of content used the reported method with slight modifications. Cholesterol content in the egg yolk emulsion was determined at 550nm by o-phthalaldehyde method in ultraviolet-visible spectrophotometer. Sodium cholate content was determined at 620nm by furfural colorimetry in U-Vis. The concentration of unsaturated oil was determined at 530nm following the vanillin phosphate chromogenic method in U-Vis. The mass concentration of heavy metal ions in the solution was determined by the atomic absorption spectrophotometry in the Z-2000 atomic absorption spectrometer.

4、In Section 2.2, more detail must be provided such as initial concentrations of the substances, total solution volume, etc.

Response: I'm really sorry that we didn't write in detail at this part. And we have made modified in manuscript. The contents of each substance and BC in each adsorption systems are listed in Table 1, and the total volume of solution is 100mL. (L110)

5、The characteristics of BC powder must be informed and used to explain along with the adsorption mechanism.

Response: BC is composed of nanoscale microfiber network structure, and its unique three-dimensional network structure provides the possibility to adsorb other substances. Meanwhile its molecules contain a large number of hydrophilic groups such as hydroxyl, which also provides the possibility to bind other substances.

We have added it in the introduction section, and marked in red. (L57-59)

6、The adsorption mechanism must be explained clearly. 

Response: In the adsorption process, the adsorbent rapidly occupies the cavity inside BC after contact with BC. With the increase of adsorption time, the adsorption site decreases, and the difference between the concentration of the adsorbent and the concentration of the sorbent becomes smaller, the driving force decreases, and finally the adsorption reaches the equilibrium state. We have described it in the manuscript (L245-252). And according to our previous studies, BC has both membrane diffusion and intra - particle diffusion, but we did not study in depth in this paper.

7、In line 226, the empty pores were mentioned. Were the substances adsorbed inside the BC pores? 

Response: Yes, adsorbents will enter the BC network empty pores and occupy empty pores, which will decrease with the increase of adsorption.

8、If the adsorption occurs only on the BC surface area, how does diffusion affect adsorption? 

Response: We are so sorry that we did not express clear in the article. The adsorption of BC to adsorbents is more than superficial, the adsorbents may also enter the network cavity inside the BC.

9、Adsorption type should be classified for each substance. 

Response: In our study, we find that the adsorption type maybe consistent. In the adsorption process, the adsorbent rapidly occupies the cavity inside BC after contact with BC. With the increase of adsorption time, the adsorption site decreases, and the difference between the concentration of the adsorbent and the concentration of the sorbent becomes smaller, the driving force decreases, and finally the adsorption reaches the equilibrium state. And we described it in the manuscript.

10、The result of the adsorption part was presented without the statistical analysis (the plot without error bars). 

Response: We are so sorry for the mistake. And we have made corresponding modifications in the manuscript.

11、The results should be compared with previous works.

Response: Thank you for your advice. Our study find that BC has blood lipid-lowering and improve the lipid metabolism disorder, which was consistent with previous studies. And we have made supplements in the conclusion part, and marked in red (L519-524). Meanwhile, previous studies have indicated that the BC has good adsorption performance, regulating blood lipid, and improve the intestinal flora as dietary fiber, but there are no system studies about BC lowering lipid, so we study the BC adsorption ability in vitro and effect on metabolic capacity of hyperlipidemia mice, and corresponding describe in the introduction section. (L59-70)

Reviewer 2 Report

The work presented by Wen Zhang and co-workers Effects of bacterial cellulose on lowering lipid levels in vitro 2 and improving lipid metabolism in vivo. In this respect the research design and methods, authors choose proper characterization techniques that lead to constructive conclusions. The manuscript is well prepared from the editorial as well as the scientific point of view. Nevertheless, I found some points that need to be improved prior to publication, see below:

Introduction:

  • When you are focus on the adsorption capacity of BC, you need to make some review to the physical, chemical, surface, and adsorption properties of BC in this part.
    - The aim, problem to be solved and novelty of the works are better to display in the introduction part.

Materials and Methods
- The BC, samples used and aqueous media were prepared by referring the other works. But there are no description in this paper. For the publication of original article, it is important to explain this part even less you made some modification.
- Line 106: … in emulsion… —— how to prepare the emulsion??
- I didn’t see the characterization method in the in vitro adsorption part.

Results
- In line 230: The van der Waals force between cholesterol and BC also gradually weakened —— put the references
- Why did the pH 2 give better result than 7?
- In line 250: During rapid adsorption, the large surface area of BC could have helped bind with sodium cholate ——put the references.
- 3.1.3 Adsorption of unsaturated oil by BC —— there is no scientific justification to describe your result.
- Line 305 – 307: The adsorption of heavy metal ions onto BC is mainly caused by the surface-active groups (hydroxyl groups)…. —— why?
- Figure 4 —— why did in heavy metal adsorption give an opposite result for both variable pH and sample (Cr6+ and Pb2+), what happen in there?

Author Response

Introduction:

1、When you are focus on the adsorption capacity of BC, you need to make some review to the physical, chemical, surface, and adsorption properties of BC in this part.

Response: BC is composed of nanoscale microfiber network structure, and its unique three-dimensional network structure provides the possibility to adsorb other substances. Its molecules contain a large number of hydrophilic groups such as hydroxyl, which also provides the possibility to bind other substances. We have added it in the introduction section, and marked in red. (L57-59) And the properties of BC have been thoroughly studied in our earlier paper.

2、The aim, problem to be solved and novelty of the works are better to display in the introduction part.

Response: In our work, aim to make out the effects of bacterial cellulose on lowering lipid levels, the BC’s abilities to reduce lipids in vitro and improve the lipid metabolism of hyper-lipidemia mice were analyzed. In vitro, we study the adsorption property of BC to some substances (cholesterol, sodium cholate, unsaturated fat and heavy metal ions) related to lipid metabolism. In vivo, BC was added at high, middle, and low doses to a high-fat diet to carry out the random intervention feeding experiment and evaluated the effects of BC on blood lipid index, liver histopathology, and intestinal microflora in hyperlipidemia mice. We have made corresponding complement in the introduction section, and marked in red. (71-79)

Meanwhile, previous studies have indicated that the BC has good adsorption performance, regulating blood lipid, and improve the intestinal flora as dietary fiber, but there are no system studies about BC lowering lipid, so the study’s findings will provide a theoretical basis for the utilization of BC in lose-fat-foods and provide certain reference meaning for healthy diet, the corresponding describe in L59-70.

Materials and Methods

3、The BC, samples used and aqueous media were prepared by referring the other works. But there are no description in this paper. For the publication of original article, it is important to explain this part even less you made some modification.

Response: Thank you for your advice. The modification has been made according to your suggestion, and the revised part has been marked in red in manuscripts(L82). And the contents of each substance and BC in each adsorption systems are listed in Table 1, and the total volume of solution is 100mL. (L110)

4、Line 106: … in emulsion… —— how to prepare the emulsion?

Response: We are so sorry for the improper use of words. It should not be “emulsion” here, but “system”, which has been modified in the manuscript. (L119 and 121).

5、I didn’t see the characterization method in the in vitro adsorption part.

Response: I'm so sorry that our manuscript was not detailed enough. We have modified in 2.2 and marked in red. The adsorption of cholesterol, sodium cholate, unsaturated oil, heavy metal ions and the determination of content used the reported method with slight modifications. Cholesterol content in the egg yolk emulsion was determined at 550nm by o-phthalaldehyde method in ultraviolet-visible spectrophotometer. Sodium cholate content was determined at 620nm by furfural colorimetry in U-Vis. The concentration of unsaturated oil was determined at 530nm following the vanillin phosphate chromogenic method in U-Vis. The mass concentration of heavy metal ions in the solution was determined by the atomic absorption spectrophotometry in the Z-2000 atomic absorption spectrometer. (L94-102)

Results

6、In line 230: The van der Waals force between cholesterol and BC also gradually weakened —— put the references

Response: Thanks for your reminding. And we have inserted relevant references. (L251)

7、Why did the pH 2 give better result than 7?

Response: The cholesterol-adsorption capacities at pH 7.0 (simulation intestinal environment) were higher than those at pH of 2.0 (simulation gastric environment). Under acidic conditions, there were more hydrogen ions, which rejected the positive charges carried by cholesterol, thereby resulting in the decreased cholesterol-binding capacity of BC. We have added it the manuscript and marked in red. (L232-238)

8、In line 250: During rapid adsorption, the large surface area of BC could have helped bind with sodium cholate ——put the references.

Response: Thanks for your reminding. And we have inserted relevant references. (L255)

9、3.1.3 Adsorption of unsaturated oil by BC —— there is no scientific justification to describe your result.

Response: We have added corresponding references to illustrate our findings, and marked in red in 3.1.3. (276-279)

10、Line 305 – 307: The adsorption of heavy metal ions onto BC is mainly caused by the surface-active groups (hydroxyl groups)…. —— why?

Response: The mechanism involved surface energy adsorption, electrostatic attraction, and coordination adsorption, and the hydroxyl groups have different protonation degree which influence the adsorption capacity. And this has been explained in detail in the manuscript. (L323-340)

11、Figure 4 —— why did in heavy metal adsorption give an opposite result for both variable pH and sample (Cr6+ and Pb2+), what happen in there?

Response: Under low pH, hydroxyl groups easily bind with H+ to perform protonation and generate -OH2+, which combines with heavy metals such as Cr6+. With increase in pH, part of Cr2O72- gets converted to CrO42-, leading to the decrease in Cr6+; therefore, BC’s adsorption quantity decreased. Under extremely low pH, Pb2+ and H+ compete for the adsorption sites on BC, and limited protonation occur. This leads to an increase in the electrostatic repulsion force on BC surface; therefore, the amount of Pb2+ adsorbed onto BC decreased. With an increase in pH, the hydroxyl groups on the BC surface dehydrogenate, resulting in an increase in negative charges. Thus, the adsorption quantity gradually increased. When pH continued to increase, the removal rate of Pb2+ further increased, probably due to the precipitation of metal hydroxides in the solution, instead of the increase in adsorption by BC. This has been explained in the manuscript with references. (L330-340)

Reviewer 3 Report

The manuscript reports the use of bacterial cellulose in adsorption of various substances and the effect of this material onto improving lipid metabolism in vivo. The manuscript is interesting, however the Authors should addresses the following comments to improve the manuscript:

- ”Unsaturated oil” phrase placed in abstract should be explained;

- “Cr6+” is incorrect, it should be showed as actually investigated ion;

- Perchlorate is an oxidizing agent, were any signs of oxidation observed when the material was used for adsorption?;

- What are the analytical procedures used for measuring concentrations of all analytes which were “described previously”? Is should be known to the reader how were the concentrations measured. All necessary information should be placed in the experimental section;

- Adsorption isotherms should be performed along with appropriate model application (Langmuir for instance);

- Adsorption kinetic models should be used for data interpretation;

Author Response

Reviewer #3

The manuscript reports the use of bacterial cellulose in adsorption of various

substances and the effect of this material onto improving lipid metabolism in vivo. The

manuscript is interesting, however the Authors should addresses the following

comments to improve the manuscript:

1、 ”Unsaturated oil” phrase placed in abstract should be explained;

Response: Oil often indicate liquid fat. In our study, the “unsaturated oil” was soybean oil, so we used “unsaturated oil”.

2、“Cr6+” is incorrect, it should be showed as actually investigated ion;

Response: In our work, this part of the study is to explore the adsorption property of BC to heavy metal ions, so we prepare of potassium dichromate solution and BC dispersed in the solution, to study the adsorption property of BC to Cr6+ in potassium dichromate solution, among which Cr6+ is present in the potassium dichromate solution.

3、Perchlorate is an oxidizing agent, were any signs of oxidation observed when the

material was used for adsorption?

Response: We didn't use perchlorate in the experiment and there is no mention of perchlorate in the manuscript.

4、What are the analytical procedures used for measuring concentrations of all analytes

which were “described previously”? Is should be known to the reader how were the

concentrations measured. All necessary information should be placed in the

experimental section;

Response: I'm so sorry that our manuscript was not detailed enough. We have modified in 2.2 and marked in red (L94-102). The adsorption of cholesterol, sodium cholate, unsaturated oil, heavy metal ions and the determination of content used the reported method with slight modifications. Cholesterol content in the egg yolk emulsion was determined at 550nm by o-phthalaldehyde method in ultraviolet-visible spectrophotometer. Sodium cholate content was determined at 620nm by furfural colorimetry in U-Vis. The concentration of unsaturated oil was determined at 530nm following the vanillin phosphate chromogenic method in U-Vis. The mass concentration of heavy metal ions in the solution was determined by the atomic absorption spectrophotometry in the Z-2000 atomic absorption spectrometer.

5、Adsorption isotherms should be performed along with appropriate model application (Langmuir for instance);

Response: Thanks for your suggestion, but the work we have completed has not performed kinetic simulation for the adsorption process of each adsorbent, and we will complete this work in the subsequent research.

 6、Adsorption kinetic models should be used for data interpretation

Response: Thanks for your suggestion again, and we will continue to complete this work in the subsequent research.